

# 1 Multivariate hydrological data assimilation of soil moisture and
# 2 groundwater head

Donghua Zhang[1], Henrik Madsen[2], Marc E. Ridler[2], Jacob Kidmose[3], Karsten H. Jensen[1], and Jens C.
Refsgaard[3]
[1]Department of Geosciences and Natural Resource Management, University of Copenhagen, Copenhagen, Denmark
[2]DHI, Hørsholm, Denmark
[3]Geological Survey of Denmark and Greenland (GEUS), Copenhagen, Denmark
*Correspondence to*: Donghua Zhang (donghua.zhang@ign.ku.dk)
**Abstract.** We present a method to assimilate observed groundwater head and soil moisture profiles into an integrated
hydrological model. The study uses the Ensemble Transform Kalman Filter (ETKF) data assimilation method with the
MIKE SHE hydrological model code. The method was firstly tested on synthetic data in a catchment of less complexity (the
Karup catchment in Denmark), and later implemented using real data in a larger and more complex catchment (the
Ahlergaarde catchment in Denmark). In the Karup model, several experiments were designed with respect to different
observation type, ensemble size and localization scheme, to investigate the assimilation performance. The results showed the
necessity to using localization especially when assimilating both groundwater head and soil moisture. The proposed scheme
with both distance localization and variable localization was shown to be more robust and provide better results. Using the
same assimilation scheme in the Ahlergaarde model, groundwater head and soil moisture were successfully assimilated into
the model. The hydrological model with assimilation showed a much improved performance compared to the model without
assimilation.



## 1 Introduction

Integrated hydrological modelling plays an important role in water resources management to develop sustainable environmental and economic schemes. Integrated models offer advantages with respect to incorporating different physically-based hydrological processes and providing a consistent prediction of different hydrological variables. Hydrological data assimilation aims to efficiently combine both the knowledge represented by integrated hydrological modelling and the information gained from observations. Data assimilation (DA) has the advantage to utilize both imperfect models and limited observations to provide a more accurate prediction and considering uncertainties on both sides.

Groundwater head and soil moisture are two key variables in hydrological modelling of the saturated and unsaturated zones. Several applications of assimilating each variable individually in either groundwater models or land surface models have been reported. For example, (Chen and Zhang, 2006) presented an application of the Ensemble Kalman Filter (EnKF) to a groundwater flow model with updating of both groundwater head and hydraulic conductivity. (De Lannoy et al., 2007) applied EnKF for soil moisture state and bias estimation in a small field using the CLM (Community Land Model), (De Lannoy et al., 2006)). There are also few applications with assimilating both groundwater head and soil moisture. For example, (Visser et al., 2006) used groundwater head and soil moisture data to re-calibrate the SWAP (Soil, Water, Atmosphere and Plant) model on-line using a simplified form of Newtonian nudging, and showed the superior results compared to off-line calibration.

The combination of multivariate assimilation and integrated hydrological models provides great potential to deepen our understanding of the value of different measurement data. Several studies of multivariate assimilation applications in integrated hydrological models have been reported. (Xie and Zhang, 2010) applied EnKF to the Soil and Water Assessment Tool (SWAT), with updating of multiple states and parameters including runoff, soil moisture and evapotranspiration. (Camporese et al., 2009) used EnKF in the CATHY (CATchment HYdrology) model with coupled surface and subsurface flow, to assimilate groundwater head and stream discharge. (Rasmussen et al., 2015) assimilated the same variables using the ensemble transform Kalman filter (ETKF) with the MIKE SHE model. (Shi et al., 2014, 2015) employed EnKF to assimilate multivariate hydrological states in a small catchment modelled by the land surface model Flux-PIHM, with a focus on parameter estimation. (Lee et al., 2011) used the variational assimilation approach to assimilate streamflow and in-situ soil moisture, to correct the soil moisture profiles within the HL-RDHM model. (Ridler et al., 2014b) developed a generic DA framework that enables coupling hydrological models with the OpenDA library (http://www.openda.org) using the Open Model Interface OpenMI (Gregersen et al., 2007), and applied it with the MIKE SHE model. (Han et al., 2015) developed an open source multivariate DA framework DasPy for the Community Land Model. Although many multivariate DA platforms and applications have been reported, assimilating both soil moisture and groundwater head in an integrated hydrological model has not been studied in detail. Representing two important hydrological variables, their observational values by assimilation in integrated hydrological models are explored in this study.



Meanwhile, techniques have been developed for multivariate DA. The most straightforward approach used in integrated
models is state augmentation, which is commonly applied with EnKF and its variants, with nearly no additional
modifications on algorithms. The observation vector can be extended to accommodate multiple types of observations.
Similarly, the state vector can be augmented to include all relevant state variables, and possibly model parameters. The
covariance matrix is thereby expanded to a block matrix with each block the cross-covariance between variables in the state
vector. (Montzka et al., 2012). A potential challenge in this respect is that implementing EnKF techniques like localization
no longer becomes straightforward. Commonly used localization techniques usually belong to covariance localization
(Hamill et al., 2001) or local analysis (Anderson, 2003). When updating a single state variable with corresponding
measurements, distance localization is usually used to reduce the impact of long distance sampling errors in the forecast
error covariance due to a limited ensemble size. In addition, when there are more than one state variable, the degree of
localization for each variable needs to be appropriately specified. Another incidental fact in multivariate DA is that the
spurious correlation across variables is usually more pronounced leading to deterioration of the model updating. To
overcome this problem, (Kang et al., 2011) successfully introduced 'variable localization' in addition to distance localization
and tested this with the local ensemble transform Kalman filter (LETKF) in a carbon cycle model.
In this study, we systematically investigate the performance of assimilating soil moisture and groundwater head, with respect
to the assimilated variable type, localization scheme and ensemble size. The assimilation method is based on ETKF (Sakov
et al., 2010), distance localization using local analysis (Sakov and Bertino, 2010), and variable localization (Kang et al.,
2011). The approach is first tested on a catchment of less complexity (the Karup catchment in Denmark) and using
synthetically generated data, and later implemented in a larger and more complex catchment (the Ahlergaarde catchment in
Denmark) using real data. From the methodology point of view, the novelty of this study is the use of advanced multivariate
assimilation methodologies in combination with application of different localization schemes. From the application point of
view, the novelty of this study is to investigate the value of assimilated variables and their impact on other processes through
integrated hydrological modelling in a complex catchment using real data.
The paper is organized as follows: the two study areas and the hydrological modelling processes are introduced in section 2;
the detailed assimilation methodology is described in section 3; section 4 presents the experimental settings and the
assimilation results based on the Karup catchment; section 5 presents the real observations, experimental settings and the
results based on the Ahlergaarde catchment; and finally general discussions and conclusions are given in section 6.
**2 Hydrological Modelling**
**2.1 Study areas**
Two study areas in Denmark are used in this study. The 440 km$^2$ Karup catchment is located in the centre of Jutland (left in
Fig. 1). The land use is mainly agriculture, and topographical elevation is between 20 and 100 m. The catchment lies in an
alluvial plain with coarse sandy soils and a strongly groundwater dominated hydrological regime. The Ahlergaarde



catchment is located in one of the most irrigated areas of Denmark (right in Fig. 1). Of the total catchment area of 1044 km$^2$,
61% is covered by agricultural crops. The surface geology consists mostly of sand and also in this catchment the streamflow
is dominated by groundwater inflow.
The Karup catchment is a well-studied catchment (Refsgaard, 1997;Madsen, 2003;Zhang et al., 2015). A relative simple
model with fast computation time was developed for this catchment to test and verify various DA methods. The Ahlergaarde
catchment is the research catchment of the Danish Hydrological Observatory (HOBE) (Jensen and Illangasekare, 2011). This
study area is ideal to further test DA methods using real measurements.
**2.2 Hydrological model**
The MIKE SHE hydrological modelling system is used for developing models for the above two catchments. As a
physically-based distributed hydrological model, MIKE SHE simulates the major processes in the water cycle including
evapotranspiration, overland flow, unsaturated flow, groundwater flow, river flow and the interactions between them. MIKE
SHE also has the flexibility of modelling each process at given spatial and temporal resolutions with different complexity.
The complexity can be chosen according to the model purpose and data availability (Graham and Butts, 2005).
In the Karup catchment, the modelling is based on the following process descriptions: groundwater flow is assumed
horizontal and is therefore modelled by one computational layer, drain flow (pipes/ditches) is described by a simple
conceptual relationship and occurs when the groundwater table exceeds the drain level, 1D unsaturated flow is assumed and
based on a simplified gravity-based flow equation, 1D channel flow is assumed and based on kinematic routing, 2D overland
flow routing is based on the diffusive wave approximation of the Saint Venant equations, evapotranspiration is described
including interception, soil evaporation and transpiration by vegetation. The numerical discretization in the horizontal plane
is 1000 x 1000 m$^2$ grid size. The model is forced by station-based daily precipitation and uniform daily values for reference
evaporation.
For the Ahlergaarde catchment, the same components are included as for the Karup catchment. For computational efficiency,
irrigation is not included. The modelling approaches are the same as for Karup except that 3D groundwater flow is
considered with six numerical layers defined according to geological stratigraphy. Another main difference is that the model
uses a smaller grid size (200 x 200 m$^2$). The finer model discretisation enables the model to utilize the finer resolution
system data such as geological stratigraphy, soil type and land use. The model is forced with grid-based daily precipitation
(10 km), temperature (20 km) and reference evaporation (20 km) from the Danish Metrological Institute. In both catchments
no-flow boundaries are defined along the catchment borders. The model parameterisation and model calibration are
introduced in the next section.
The improved model resolution and complexity in the Ahlergaarde catchment increased the simulation time significantly.
For example, the average model time step length in groundwater zone decrease from 7.5 hours in the Karup model to 1.3
hours in the Ahlergaarde model. In consequence one year's model simulation takes less than one minute for Karup and





around one hour for Ahlergaarde. The differences in model resolution and simulation time for the two catchments are
summarized in Table 1.
**2.3 Model calibration**
For both catchments, the model parameterisation is kept relatively simple yet able to represent the overall spatial patterns of
key model parameters. When specifying the parameter values for each property class (e.g., geological unit, vegetation types
and soil types), most of the parameters cannot be estimated empirically or directly inferred from the data. Thus model
calibration is usually required using an optimization algorithm like AUTOCAL (Madsen, 2003) or PEST (Doherty, 2010).
For the Karup model, the most sensitive parameters describing the hydraulic properties of the river, unsaturated zone,
saturated zone, and river-aquifer interaction are calibrated using AUTOCAL (Zhang et al., 2015). As calibration data we use
35 biweekly groundwater head observations and daily observations of stream discharge for a six year period (1969-1974)
(Fig. 1).
The Ahlergaarde model is calibrated using PEST version 11.8 (Doherty, 2010). The data used in the calibration are
groundwater head observations (466 in total) scattered over the catchment (not shown in Fig. 1) and river discharge
observations from the period of 2006 - 2009. In most of the groundwater wells only one observation is available for the
entire calibration period and only few wells have time series. Discharge data comprise time series of daily values from five
stations (Fig. 1). The original calibrated model uses a simplified two-layer approach to simulate unsaturated flow and
evapotranspiration, where the average soil moisture is calculated for the root zone and the layer below the root zone. In order
to assimilate in-situ soil moisture data at different depths, the gravity flow module is used as a replacement of the two-layer
approach in the unsaturated zone. By doing so, the soil moisture is calculated using a 200 m horizontal grid resolution at
different depths. The overall modelling performance in terms of water balance and discharge dynamics becomes marginally
reduced compared to the calibration results obtained.
**3 Data assimilation**
**3.1 Ensemble transform Kalman filter**
The assimilation algorithm used in this study is the ETKF, which is a popular variation of the EnKF (Evensen, 2003).
Similar to EnKF, ETKF is a Monto Carlo implementation of the Kalman filter, which approximates the posterior probability
distribution conditioned on a series of observations, and is able to deal with nonlinear models. However, different from
EnKF, ETKF is a deterministic filter as it does not require additional observation perturbations, which might introduce
sampling noise. The ETKF was originally introduced by (Bishop et al., 2001), and later modified to be unbiased (Wang et al.,
2004). As an ensemble-based deterministic filter, it has the advantage to calculate the forecast error covariance efficiently. It
is also computationally faster than the ensemble square root filter (EnSRF, (Whitaker and Hamill, 2002)).
To develop the DA algorithm, a state-space formulation is needed


$$X_{t+1} = M(X_t, U_t, \theta) \approx M_d\big(X_t, \widetilde{U_t}, \tilde{\theta}\big) \tag{1}$$

where $M$ is the stochastic model operator based on the numerical solution to the MIKE SHE equations, $M_d$ is the
deterministic MIKE SHE model operator, $X_t$ and $U_t$ are the state vector and model forcing at time step $t$, $\theta$ stands for the
model parameters. $\widetilde{U_t}$ and $\tilde{\theta}$ are the perturbed forcing and parameters respectively. Note that the stochastic model operator $M$
is approximated by the deterministic MIKE SHE model with taking both model forcing uncertainty and model parameter
uncertainty into account (Zhang et al., 2015). In both models, precipitation and potential evapotranspiration are perturbed by
adding a random Gaussian noise to the actual value. The parameter uncertainty is described using the covariance estimated
from calibration. The selected parameters are assumed to be multivariate normal distributed and perturbed using Latin
hypercube sampling based on the associated parameter covariance. Additional post-processing steps are used to ensure that
the perturbed parameters are still within realistic parameter ranges.
At time $t +1$, the observations can be written as,

$$Y_{t+1} = HX_{t+1} + \varepsilon_{t+1}, \varepsilon_{t+1} \sim N(0, R_{t+1}) \tag{2}$$

where $Y$ denotes the observation vector, and $H$ is the linear mapping operator specifying the deterministic relationship
between observations and model state $X$. In this study, the observations are either groundwater head, soil moisture or both.
Similarly, the state vector consists of groundwater head, soil moisture, or both. When two variables are assimilated, the state
vector is augmented to accommodate both variables at all computational cells, and the observation operator $H$ is revised to
select the correct model equivalent and compare with the corresponding observation. The observation noise is assumed to be
Gaussian, temporally-uncorrelated, spatially-uncorrelated, with zero-mean and a prescribed constant standard deviation $\sigma_r$.
Therefore, $R_{t+1}$ is a diagonal matrix with constant values along the diagonal (i.e., $R_{t+1} = diag(\sigma_r^2, ..., \sigma_r^2)$).
The forecast state distribution can be estimated by a finite number $m$ of model realizations from Eq.(1) as follows,

$$X^f = [x^{f1}, x^{f2}, ..., x^{fm}] \tag{3}$$

where the superscript $f$ stands for 'forecast'.
The forecast error covariance can be written as

$$P^f = X'^f (X'^f)^T / (m-1) \tag{4}$$

where $X'^f$ is the forecast ensemble perturbation

$$X'^f = [x^{f1} - \overline{X^f}, x^{f2} - \overline{X^f}, ..., x^{fm} - \overline{X^f}] \tag{5}$$

and $\overline{X^f}$ is the ensemble mean. After assimilation, both the analysed state mean and the analysed error covariance can be
calculated:

$$\overline{X^a} = \overline{X^f} + K(Y - \overline{HX^f}) \tag{6}$$

$$P^a = (I - KH)P^f \tag{7}$$

where the superscript $a$ stands for 'analysed', and $K$ is the Kalman gain defined as

$$K = P^f H^T (H P^f H^T + R)^{-1} \tag{8}$$





In practise, $P^a$ is never explicitly calculated and only the ensemble mean and ensemble anomalies are updated. Based on
factorizing Eq. (7) on both sides of the equation the following equation is obtained:

$$X'^a = X'^f T \qquad (9)$$

where

$$T = [I + (HX'^f)^T R^{-1} HX'^f /(m-1)]^{-1/2} U \qquad (10)$$

and $U$ is an arbitrary orthonormal matrix $UU^T = I$.
The MIKE SHE model is coupled with a generic DA library that handles the time propagation and update of the model
ensemble based on the ETKF (Ridler et al., 2014b).

## 3.2 Localization

In ensemble based Kalman filter systems, the true state and the uncertainty are represented by a limited ensemble of
realizations. The undersampling can lead to filter inbreeding and spurious correlations in the error covariance matrix, which
potentially can lead to filter divergence. Localization is a commonly used technique when applying ensemble based Kalman
filters to overcome this problem. By artificially reducing the impacted spatial domain of observations, the spurious
correlation between two remote locations can be avoided. For each element in the state vector, local analysis (LA, (Sakov
and Bertino, 2010)) is used to approximate the state error covariance within the local window. The ensemble anomalies
outside this local window will be unchanged during the filter updates. However, LA is usually applied to a single state
variable for which certain spatial correlations exists. When the state vector contains two or more variables, specifying the
localization degree for each variable is not straightforward. More importantly, correlations between variables are not clear
because physical distances between variables may not exist. Similar to the approach by (Kang et al., 2011), we introduced
different variable localization schemes based on whether the correction of one variable can impact the update of other
variables. In this section, the distance localization will be introduced first followed by the variable localization.

## 3.2.1 Distance localization

We formulate the distance-localized ETKF equations with similar notations as in (Sakov and Bertino, 2010). A variable with
an upper accent '$i$' means a local variable, which is used to update the $i$'th element of the state vector. During the updating
with localization, $i$ is looped for each element in the state vector. For example, $\overset{i}{K}$ means the local Kalman gain, $\overset{i}{y}$ denotes the
local observations associated with the $i$'th element in the state vector. In matrices, the subscript '$i,:$' refers to the $i$'th row. To
avoid the occasional sudden changes of analysis from one state vector element to the next one when an observation just
arrives or exits the local window, an ensemble tapering with a distance-based tapering function $f(.)$ is used to ensure the
impact of the observation is reduced gradually from the centre to the boundary within the local domain (Sakov and Bertino,

28 2010).

Therefore, to update the $i$'th element, the localized-ETKF equations (Eq. (6), (9), (10)) become





$$\overline{X_i^a} = \overline{X_i^f} + \overset{i}{K_{i,:}}(\overline{Y - \overset{i}{H}\overset{i}{X^f}}) \tag{11}$$

$$\overset{i}{K_{i,:}} = \overset{i}{X_{i,:}'^f}\overset{i}{S^T}(I + \overset{i}{S}\overset{i}{S^T})^{-1}\overset{i}{R^{-1/2}}/\sqrt{m-1} \tag{12}$$

$$\overset{i}{X_{i,:}'^a} = \overset{i}{X_{i,:}'^f}\overset{i}{T} \tag{13}$$

$$\overset{i}{T} = (I + \overset{i}{S}\overset{i}{S^T})^{-1/2}U \tag{14}$$

$$\overset{i}{S} \overset{\text{def}}{=} \overset{i}{R^{-1/2}}\overset{i}{H}\overset{i}{X'^f}/\sqrt{m-1} \tag{15}$$

During the update, the observation $Y$, innovations $Y - HX^f$, observation error variance $R$ and ensemble observation
anomalies $HX'^f$ are tapered in line with the taper function *f(.)*. The LA taper function is usually determined by the distance
between two model points, which decreases from one to zero as the distance increases. Different choices of distance-
dependant covariance functions can be used according to dimension and physical property. For example, (Sakov and
Bertino, 2010) use the Gaspari and Cohnv 1D taper function to compare different localization methods. (Ridler et al., 2014a)
use a 2D squared exponential covariance function as taper function to localize the soil moisture updating. In this study, due
to the difference in variable type and variable dimension, the taper function is chosen to be case specific based on the 2D
squared exponential covariance function.
For groundwater heads, in both catchments, the LA taper function is chosen to have a radius of 5 km, to include a relatively
large number of observations to correct each node, and also to provide larger spatial influence of the update. For the
Ahlergaarde catchment where the groundwater is modelled in 3D, the LA localization is applied to each layer with the same
radius.
For soil moisture, the measurements usually represent a relatively smaller spatial scale. In both catchments, localization
scales are specified to ensure that the state correction from the assimilated observation is localized. Horizontally, the taper
function is chosen to have a radius of $1 - 5$ km at the layer where soil moisture is screened. Because most of the data are
measured in the surface and near-surface soil (5 - 25 cm depth), the water content in the upper layers (e.g., within 1 m or 0.5
m depth) are expected to have a larger correction compared to the water content in deeper layers. Therefore, when the depth
exceeds the soil moisture observation, we add a quadratically increasing cut-off value for the covariance function as the
depth increases (Fig. 2).
**3.2.2 Variable localization**
Variable localization is an option when assimilating both groundwater head and soil moisture. Variable localization
determines whether the information from one variable can be used to update the other. When variable localization is off, no
matter the available observation type (groundwater head, soil moisture or both), all observation data are used to update the
ensemble mean (Eq. (11)) and anomaly (Eq. (13)) for both variables. Therefore the correlation between the variables is kept





during the assimilation. In addition, if distance localization is applied, the correlation exists in localized domains between
variables.
When variable localization is applied, each observation type will only be used to update its own type of state variable. Other
variables in the state vector will be unchanged during update. If distance localization is applied, state updates are spatially
localized within its own type of variable.
Practically, the variable localization can be done by slight modifications to Eq. (11-15). The tapering function is extended to
have an 'if/else' statement prior to the existing distance-based tapering function, depending on variable localization is chosen
or not. Here we explain the process of updating one element when variable localization is applied. When looping over the
$i$'th element in the state vector, the state in the 'local' window is selected first by ensuring it has the same variable type as in
the $i$'th element, then calculating the weight according to the distance from the $i$'th element. For example, when updating soil
moisture in a grid cell, the ensemble mean and anomaly will be unaffected by soil moisture observations outside the local
window, as well as by groundwater head observations.

## 4 Study in the Karup catchment

In the Karup catchment experiment, the calibrated model described in section 2.3 is used as the deterministic model. The
ensemble is generated by adding appropriate model error to the deterministic model. Similarly, given the predefined model
error, a single random model realisation is generated to be the "true" model. Note that the "true" model here is only an
assumption of reality. The model error is defined by perturbing both model forcing (precipitation and potential
evapotranspiration) and selected model parameters (Zhang et al., 2015). The ensemble is running freely from 1969/12/01 to
1973/01/01 as a warm-up period. During the warm-up period, each ensemble member starts with the same initial condition
but has different model trajectories because of different forcing and parameter values. It is important to generate an ensemble
with a realistically large spread, so that the model uncertainty can be fully represented by the ensemble spread.
The synthetic observation used to be assimilated are generated from the "true" model. Given the true realization, by adding
measurement errors to certain model variables at given time and location, a set of synthetic observations can be produced.
Both groundwater head and soil moisture (depths of 5 cm and 25 cm) are extracted from the same 35 locations as the actual
head observations (Fig. 1). For simplicity, the observation noise for each variable is assumed to be white Gaussian, with
homogeneous and constant standard deviation of 0.15 m for head and 5% for the soil volumetric water content. Due to the
fact that groundwater head has a much slower dynamic compared to the unsaturated flow, we assimilate head with weekly
frequency and soil moisture with daily frequency.
After the warm-up period, the synthetic observations are assimilated over a one year period from 1973/01/01 to 1974/01/01.
Given the fact that the "true" model is known, the deterministic model can be seen as an imperfect model. With the purpose
to combine the imperfect model and the synthetic observations, different experiments are carried out to investigate under
which condition the assimilation result are most similar to the 'true' model. These experiments are designed using different





observation variables, localization scheme and ensemble size. The assimilation performance can be assessed by taking the
root mean square error (RMSE) between the model simulation and the true state for selected variables over the entire domain
at all available time steps. As soil moisture measurements are depth-dependent, RMSE is calculated for each depth (each
layer). Here we not only show the results from 5 cm and 25 cm depths where observations are assimilated, but also at 50 cm
depth. In addition, other hydrological responses to the assimilation (e.g., evapotranspiration and discharge) are evaluated.

## 4.1 Univariate assimilation

When a single variable is assimilated (groundwater head or soil moisture), the state vector only consists of the corresponding
observed variable at all model grid cells. Therefore, the remaining variables will not be changed directly from the filter.
However, as both the groundwater component and unsaturated zone are fully coupled with surface water and other model
components, the whole model state will be affected from updating a single variable. Different experiments are carried out
using an ensemble size of 60:
NoDA: deterministic model without DA.
DA_H: assimilating head without localization.
DA_HLoc: assimilating head with horizontal localization radius of 5 km.
DA_SM5: assimilating soil moisture at 5 cm depth without localization.
DA_SM5Loc: assimilating soil moisture at 5 cm depth with localization of 5 km spatial radius and 1 m depth.
DA_SM5LocSmall: assimilating soil moisture at 5 cm depth with localization of 3 km spatial radius and 50 cm depth.
DA_SMBothLoc: assimilating soil moisture at both 5 cm and 25 cm depths with 5 km spatial radius and 1 m depth.
As the experiment names indicate, H stands for groundwater head and SM stands for soil moisture. Loc indicates that
localization is added to the experiment.
Results from the DA experiments are shown in Fig. 3. When head is assimilated (DA_H), RMSE for head improves
significantly from 0.21 m to 0.08 m. However, the soil moistures at the three depths are basically not influenced, with similar
performance at 5 cm and 25 cm, and a slight deterioration at 50 cm. When localization is used (DA_HLoc), the corrections
are localized around the head observations and the overall performance is slightly degraded.
When soil moisture at 5 cm depth is assimilated alone without localization (DA_SM5), the soil moisture profile clearly
improves at all three depths. However, for head the performance is almost the same as in the deterministic model. Different
localization scales have been tested with assimilating soil moisture at 5 cm depth (DA_SM5Loc and DA_SM5LocSmall).
The result indicates that the overall assimilation performance decreases with smaller localization scale.
As we can see from the above, the experiments with assimilating head or soil moisture show degraded results using
localization. This could be explained as follows. Firstly, spatial correlations are affected by the catchment size and the
relatively large grid size used. Pronounced correlations exist even between remote locations, and therefore localization may



cut off true correlations, which leads to a worse result overall. Secondly, there are a relatively large number of observations
compared to the size of the state vector, which reduces the problem of spurious correlation. Study shows that there is a
strong relationship between the significance of spurious correlation and the number of observations (Rasmussen et al., 2015).
Localization is more effective to reduce spurious correlation when the number of observations is relatively small.
Similar to DA_SM5Loc, the experiment DA_SMBothLoc uses the same settings but assimilates soil moisture observations
at both 5 cm and 25 cm. Compared to DA_SM5Loc, the results show some improvements at 25 cm and 50 cm. Again,
groundwater head is hardly influenced by assimilating soil moisture. In the following experiments, we include observations
at both 5 cm and 25 cm when soil moisture is assimilated.
**4.2 Multivariate assimilation**
In this section, several experiments assimilating both groundwater head and soil moisture are carried out with a focus to test
different localization schemes. The abbreviation D and V indicate distance localization and variable localization
respectively.
DA_HSM: assimilating both head and soil moisture (at both 5 cm and 25 cm depths) without localization to any variable.
DA_HSMLoc_DV: assimilating both head and soil moisture (at both 5 cm and 25 cm depths) with variable localization and
with distance localization applied to head (same as DA_HLoc) and soil moisture (same as DA_SMBothLoc).
DA_HSMLoc_D: assimilating both head and soil moisture (at both 5 cm and 25 cm depths) without variable localization,
but with distance localization applied to head (same as DA_HLoc) and soil moisture (same as DA_SMBothLoc).
DA_HSMLoc_V: assimilating both head and soil moisture (at both 5 cm and 25 cm depths) with variable localization, but
without distance localization to any variable.
Results from the DA experiments are shown in Fig. 4. When neither distance localization nor variable localization is used,
all observations are used to update the state in all grids for each variable (DA_HSM). In this case the estimated correlations
between groundwater head and soil moisture are used in the update. The DA results show improved performance for soil
moisture at 5 cm and 25 cm, but much worse performance at 50 cm as well as for groundwater head. In the current filter
settings the full state covariance matrix contains unrealistic, spurious correlations, which eventually degrade the update in
the deeper soil layers.
In experiment DA_HSMLoc_DV, both distance localization and variable localization are used. Therefore, the state updates
are spatially localized within own variable and the correlation between the two variables is neglected. Particularly in this
case, when there is only soil moisture observation assimilated, the updates are limited to upper 1 m soil moisture profile
while no correction is made for head. When both types of observation are assimilated, the corrections are made for each
variable using its own error information. We can see from Fig. 4 that the experiment shows overall improved result.





In experiment DA_HSMLoc_D, distance localization is applied to head and soil moisture but variable localization is not
included. In this case, regardless of observation type, the soil moisture is corrected within 1 m depth together with head. The
result from this experiment shows improved estimate for soil moisture at 5 cm and 25 cm, together with groundwater head.
However, the soil moisture at 50 cm is slightly worsened. This indicates that the correlation between surface soil moisture
and groundwater head estimated from the ensemble is valid and improves the assimilation performance. Compared to
DA_HSM, the result shows that excluding the error information from deeper soils (below 1 m to saturation) reduces spurious
correlations and improves the performance. However, compared to DA_HSMLoc_DV, the result is slightly worse for head
and deeper soil moisture.
In experiment DA_HSMLoc_V, distance localization is off and variable localization is applied. This means that the error
information from one variable is used to update the entire domain of its own variable but does not affect the other variable.
The result indicates worse assimilation performance for soil moisture at 50 cm and for groundwater head. One potential
reason is that the lower layers of the unsaturated zone are usually fully saturated but in this experiment corrected by the
surface soil moisture observation, while the groundwater head is corrected by the head observation. Potential inconsistencies
may exist with these two updates.

## 4.3 Different ensemble size

As mentioned in section 3.2, localization allows the ensemble filters to work properly with limited ensemble size. Some of
the experiments are repeated for ensemble sizes of 30 and 90, respectively, to analyse how the assimilation performance and
the choice of localization are affected by the ensemble size. The results are shown in Fig.5.
As can be seen from Fig. 5, in the experiment assimilating head without localization (DA_H), increasing the ensemble size
(from 30 to 90) improves the head estimation. However, the performance difference between ensemble size of 60 and 90 is
small. When localization is used, the performances with all ensemble size are very similar (DA_Hloc).
In the experiment assimilating soil moisture at 5 cm depth without localization (DA_SM5), increasing the ensemble size also
improves the soil moisture in deeper depths. This indicates that using only an ensemble size of 30 introduces spurious
correlation between surface soil and deeper soil, which is reduced with larger ensemble sizes. An ensemble size of 30 also
leads to a much worse result for groundwater head compared to ensemble sizes of 60 or 90. When localization is used
(DA_SM5Loc), the assimilation performance is similar using the three ensemble sizes. Compared to the DA_SM5, there is a
large improvement in groundwater head when using ensemble size of 30.
When both soil moisture (at 5 cm and 25 cm depths) and head are assimilated without localization (DA_HSM), the
performance is generally improved when increasing ensemble size. However, increasing the ensemble size to 90 still leads to
a worse performance for soil moisture at 50 cm and groundwater head compared to the deterministic model. When
localization is used (DA_HSMLoc_DV), the soil moisture at 50 cm and the head improves as the ensemble size increases.
Overall, the assimilation performance increases in DA_HSMLoc_DV when increasing the ensemble size.



## 4.4 Actual evapotranspiration and discharge

Using an integrated model where the various hydrological processes are coupled, assimilation of head and soil moisture may also affect other model variables. The effects on evapotranspiration and river discharge are examined in this section. For actual evapotranspiration, we calculate the accumulated value over the catchment during the DA period; while for discharge, the performance at catchment outlet for the entire assimilation period is evaluated using the coefficient of determination and Nash–Sutcliffe efficiency. The results are summarized in Table 2.

The differences in accumulated actual evapotranspiration among all experiments are small (varies between 444 mm and 457 mm). When H is assimilated alone (DA_H_Loc), actual evapotranspiration is basically unchanged; while when soil moisture is assimilated, actual evapotranspiration is reduced by 7-13 mm compared to the deterministic model, as a result of correcting surface soil moisture.

The performance of discharge is slightly improved by assimilating head (DA_H_Loc). The improvement is mainly on the low flow, which is underestimated by the deterministic model. This is expected as the baseflow is determined by groundwater inflow. When soil moisture is assimilated (DA_SM5Loc and DA_SMBothLoc), the discharge also performs slightly better. However, when both variables are assimilated without localization (DA_HSM), the discharge performs significantly worse with unrealistic peak flows during spring. This is a result of the worse head estimations in the entire domain. When localization is used for soil moisture and groundwater head (DA_HSMLoc_DV), discharge is improved significantly and comparable with the deterministic model. This also demonstrates the necessity to use localization to constrain the spatial updates.

## 5 Study in the Ahlergaarde catchment

For the Ahlergaarde catchment, we use the calibrated model to simulate a 20-year period from 1990 to 2010 to provide initial conditions for the experiment used in this study. Starting from 2010-01-01, the experiment is spilt into two periods: a warm-up period (2010-01-01 to 2012-11-01) and a DA period (2012-11-01 to 2013-12-31). Similar to the approach in the Karup catchment, each ensemble member shares the same initial condition and is forced with perturbed precipitation, potential evapotranspiration and parameter values during the warm-up period. The parameter perturbation is based on parameter uncertainty information from the model calibration. Overall, we try to keep the ensemble spread relatively large and model responses physically realistic.

The deterministic model in this study, although based on a model calibrated against older data at different sites, has good skills after 2012. The model performance in terms of the hydrograph at catchment outlet in year 2013 is shown in Fig. 6, with an $R^2$ of 0.71 and Nash–Sutcliffe efficiency of 0.67. From the hydrograph, it can be seen that the model is good at predicting low flow, but sometimes overestimate the peak discharges.



## 5.1 Observations

Groundwater head are measured bi-hourly in nine wells (Fig. 1) using Eijkelkamp mini divers. The divers were installed in these wells November 2012 and thus the length of the time-series is limited. Moreover, due to occasional instrument failure the data coverages are further constrained and vary among the wells. In the groundwater model six numerical layers are defined (layer 1 in the bottom and layer 6 in the top). The nine wells are screened at different depths below the surface layer. Well M5398, M5637, M5353, and L8008 are screened in layer 5 while well M5373, M5647, M5844, M5393 and M5366 are screened in layer 4. When comparing in-situ head measurements with model predicted equivalents, large level differences usually occur due to scale disparities, and sometimes also accompanied by dynamic differences. Therefore, we calculated the average difference between observations and model simulations, and subtracted this difference from the original data. By doing so, we can avoid introducing observation bias in the assimilation system. An example of the processed observations and the open loop ensemble for well 5737 (2012-11-01 to 2013-12-31) is shown in Fig. 7 (top panel).

Soil moisture is measured at 30 sites across the catchment according to representative combinations of topography, land cover, and soil type using Decagon 5TE sensors. The dominant land uses are heath, agriculture and forest. At each site, sensors are installed at three depths: 2.5, 22.5 and 52.5 cm corresponding to measurement depth intervals of 0–5, 20–25 and 50–55 cm. Measurements are taken with 30 minutes intervals.

Most of the agriculture sites are irrigated in May and June, and the soil moisture is greatly influenced with several sudden increases during that period. However, in the model irrigation is not considered because detailed information on irrigation at the local sites is not available. Therefore, the sites where irrigation is evident from the soil moisture recordings are excluded for assimilation. In addition, a quality control to correct for systematic biases and to filter out unrealistic values has been carried out for the remaining sites. Although measurements are carried out at three depths at each site, we only use measurements at 2.5 cm and 22.5 cm depths for assimilation, as the surface/near-surface moisture is of most importance for the exchange of water and energy between land and the atmosphere. After processing, 18 out of 30 sites are used for assimilation (Fig. 1). As an example, Fig. 7 (middle and bottom panels) shows the processed soil moisture observations and the open loop ensemble at site nw1.1 (2012-11-01 to 2013-12-31).

## 5.2 Experiment settings

Similar to the experiment settings in the Karup catchment, the observation noise for each variable is assumed to be white Gaussian, with homogeneous and constant standard deviation of 0.2 m for head and 5% for the soil volumetric water content. The head and soil moisture data are interpolated to weekly and daily frequencies, respectively, for assimilation. Due to reduced computational time step and fine model resolution, the computational time for the Ahlergaarde catchment is substantial. This implies that a larger ensemble size is unaffordable. Furthermore, the more frequent data assimilation contributes to longer simulation time. From these considerations, an ensemble size of 50 is adopted. With a one year assimilation period, the simulation time is around 3-7 days depending on the experiment settings.





With the purpose of assimilating head and soil moisture, different experiments have been carried out to investigate the
assimilation performance. Considering the large model domain and fine grid, localization becomes more important here than
in the previous example. Distance localization is added to both variables separately, and variable localization is used when
both variables are assimilated. For groundwater head, we allow for update in all layers over the vertical. Horizontally, we use
a localization radius of 5 km for all layers. For soil moisture, we use the same distance localization scheme as in the Karup
catchment with a horizontal localization radius of 1 km and a vertical localization depth of 0.5m (top eight layers in the
unsaturated zone). The following experiments are carried out:
NoDA: deterministic model without DA.
DA_HLoc: assimilating groundwater head with distance localization.
DA_SMLoc: assimilating soil moisture (at both 2.5 cm and 22.5 cm depths) with distance localization.
DA_HSMLoc_DV: assimilating both groundwater head and soil moisture (at both 2.5 cm and 22.5 cm depths) with variable
localization and distance localization.

## 14 5.3 Groundwater head and soil moisture

The assimilation performance is evaluated by comparing model output with the actual observations using average RMSE.
The result is summarized in Table 3. In the experiment with assimilating head only (DA_HLoc), RMSE of head reduces
from 0.34 m to 0.21 m. However, the soil moisture predictions at both depths do not improve compared to the deterministic
model. In the experiment with assimilating only soil moisture (DA_SMLoc), RMSE of soil moisture at both depths reduces,
especially at depth 22.5cm. The head estimate, however, shows a similar performance as the deterministic model. When both
variables are assimilated (DA_HSMLoc_DV), RMSE of head reduces from 0.34 m to 0.21 m. RMSE of soil moisture
reduces from 0.044 $m^3/m^3$ to 0.040 $m^3/m^3$ at 2.5 cm depth, and from 0.034 $m^3/m^3$ to 0.028 $m^3/m^3$ at 22.5 cm depth.
Figure 8 shows the assimilated results for the same sites as shown in Fig. 7. Clearly, after 2012-11-01 when the DA period
starts, the ensemble mean is approaching the observations, especially for the head and soil moisture at 22.5 cm depth.
Although limited observations are assimilated, corrections are made for a large area within the model domain. Figure 9
shows spatial RMSEs of soil moisture and head at corresponding observation layers between the assimilation result and the
deterministic model. For each grid cell, the variables' time series values from the assimilated model and the deterministic
model are used to calculate the average RMSE.
From Fig. 9, we can clearly see the effect of the assimilation in model domain. For soil moisture, relatively large corrections
are made in 22.5cm depth compared to the surface layer. Compared to the groundwater head, however, the soil moisture
corrections are more localized. For both soil moisture and groundwater head, most of the large corrections are made at places
near the locations of observations. For groundwater head in the west and south-east regions where no head observations are
available, the corrections are generally small.





## 5.4 Actual evapotranspiration and discharge

In this section, the effect of assimilation on actual evapotranspiration and river discharge is evaluated by comparing model predictions and observations. Figure 10 compares evapotranspiration at Voulund station and discharge at the catchment outlet from the different experiments with observations. Voulund station is located in the central-north part of the catchment with several soil moisture stations around. In both graphs in Fig. 10, only small differences are seen between different simulations. Performance measures are shown in Table 4.

As shown in Table 4 the accumulated actual evapotranspiration are the same in all three assimilation experiments. However, in terms of correlation ($R^2$) some improvements are seen when soil moisture is assimilated (DA_SMLoc and DA_HSMLoc_DV). Also, there is a small improvement for discharge (Nash–Sutcliffe efficiency score) when head is assimilated (DA_HLoc). The Experiment DA_HSMLoc_DV with both variables being assimilated provides better results overall.

## 6 Discussion and Conclusion

This study investigated assimilation of soil moisture and groundwater head in an integrated hydrological model. To the best of our knowledge, this is the first study using EnKF-based methods to assimilate these two variables in an integrated hydrological model with real data. The method is based on ETKF with both distance and variable localization implemented. The proposed method is first explored for a catchment with synthetic data, and then applied to a complex model using real data.

The study shows relatively weak correlations between surface soil moisture and groundwater head in the model. First, the univariate assimilation improves the state of the variable being assimilated, but does not improve the other variable. This can be seen from the experiments in both catchments. Second, in multivariate assimilation, when the complete state error covariance of both variables is used for updating and spurious correlations are not cut off by localization, the filter failed to provide reasonable result. This indicates that unrealistic inter-variable and cross-variable correlations exist in the model ensemble.

A hybrid localization scheme which consists of variable localization and distance localization has been developed and implemented in the ETKF. Localization does not only provide better results, but also reduce the computational cost as only a section of the full state is used within the filter. Similar localization approaches have been reported in hydrological models with discharge involved (Li et al., 2013) as well as in other models (e.g., (Kang et al., 2011)). Other approaches to deal with the potential inter-variable spurious correlation include for example adaptive localization (Rasmussen et al., 2015), and using two iterative filters instead of one filter (Gharamti et al., 2013). The method used here proved to be suitable for assimilating both groundwater head and soil moisture in integrated hydrological models, and have potential to be generalized to deal with other processes.





The impact of assimilation on discharge and evapotranspiration is analysed in the Ahlergaarde with real measurements as
reference. Neither the discharge nor evapotranspiration were included in the filter state vector. However, through integrated
hydrological modelling, the discharge is improved when head is assimilated, and evapotranspiration is improved when soil
moisture is assimilated. Although the improvements are relatively small, we nevertheless see the benefits in other model
process results when improving groundwater head and soil moisture.
Increasing the ensemble size is beneficial in general, especially for estimating unobserved and un-localized variables. This is
because an increased ensemble size can better describe the true correlation in the state error covariance matrix. The effect of
ensemble size has also been widely reported in previous studies (e.g., (Xie and Zhang, 2010)). However, the balance
between the assimilation result and the computational cost is usually considered when choosing the appropriate ensemble
size for heavy models. This is an important issue for the Ahlergaarde model as the computational expenses here become
substantial. Due to the time and resource limitation, the choice of ensemble size for the Ahlergaarde model is not analysed in
the study, but will certainly be essential for real-time applications in future studies. In addition, the multivariable assimilation
should be extended with remote sensing soil moisture and other important hydrological variables (e.g., discharge) that are
not included in this study.
**Acknowledgements**
The study has been carried out with the support of the Danish Council for Strategic Research as part of the project
"HydroCast - Hydrological Forecasting and Data Assimilation", Contract No. 0603-00466B (http://hydrocast.dhigroup.com),
and S.C. Van Fonden. Field data are supplied by the HOBE project funded by the Villum Foundation
(http://www.hobecenter.dk).

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





1   **Table 1 Differences in model resolution and computation time between two catchments. SZ is saturated zone and UZ is**
2   **unsaturated zone, the symbol # means 'the number of'.**

| Catchment | Karup | Ahlergaarde |
|---|---|---|
| Area | 440 km$^2$ | 1044 km$^2$ |
| Grid size | 1000 x 1000 m$^2$ | 200 x 200 m$^2$ |
| #Grid cells in each layer in SZ | 522 | 26922 |
| #layers in SZ | 1 | 6 |
| #total grid cells in SZ | 522 | 161538 |
| #Grid cells in each layer in UZ | 438 | 26097 |
| #layers in UZ | 87 | 21 |
| #total grid cells in UZ | 38106 | 548037 |
| Time spent for 1 year simulation | Less than 1 minute | Around 1 hour |





2 **Table 2 The evapotranspiration measure (the accumulated actual evapotranspiration over the entire catchment during the DA**
3 **period) an the discharge measures ($R^2$ and Nash–Sutcliffe efficiency of discharge at catchment outlet during DA period) for each**
4 **experiment in Karup catchment.**

|  | Accumulated actual evapotranspiration (mm) | $R^2$ of discharge at outlet | Nash–Sutcliffe efficiency score of discharge at outlet |
|---|---|---|---|
| NoDA | 457 | 0.946 | 0.936 |
| DA_H_Loc | 456 | 0.961 | 0.955 |
| DA_SM5Loc | 450 | 0.961 | 0.941 |
| DA_SMBothLoc | 448 | 0.951 | 0.944 |
| DA_HSM | 444 | 0.531 | 0.484 |
| DA_HSMLoc_DV | 448 | 0.936 | 0.932 |





2  **Table 3 Average RMSE of head and soil moisture (2.5 cm and 22.5 cm) at observation locations for each experiment in**
3  **Ahlergaarde catchment.**

|  | Average RMSE of head (m) | Average RMSE of soil moisture at 2.5 cm ($m^3/m^3$) | Average RMSE of soil moisture at 22.5 cm ($m^3/m^3$) |
|---|---|---|---|
| NoDA | 0.34 | 0.044 | 0.034 |
| DA_HLoc | 0.21 | 0.045 | 0.037 |
| DA_SMLoc | 0.34 | 0.038 | 0.024 |
| DA_HSMLoc_DV | 0.22 | 0.040 | 0.028 |





2 **Table 4 Quantitative performance measures for evapotranspiration and discharge for the each experiment in Ahlergaarde**
3 **catchment.**

| | Accumulated actual evapotranspiration at Voulund (mm) | $R^2$ of actual evapotranspiration at Voulund | RMSE of actual evapotranspiration at Voulund (mm) | $R^2$ of discharge at outlet | Nash–Sutcliffe score of discharge at outlet |
|---|---|---|---|---|---|
| NoDA | 431 | 0.409 | 0.879 | 0.711 | 0.673 |
| DA_HLoc | 420 | 0.360 | 0.919 | 0.714 | 0.690 |
| DA_SMLoc | 421 | 0.452 | 0.853 | 0.704 | 0.677 |
| DA_HSMLoc_DV | 421 | 0.458 | 0.850 | 0.714 | 0.691 |





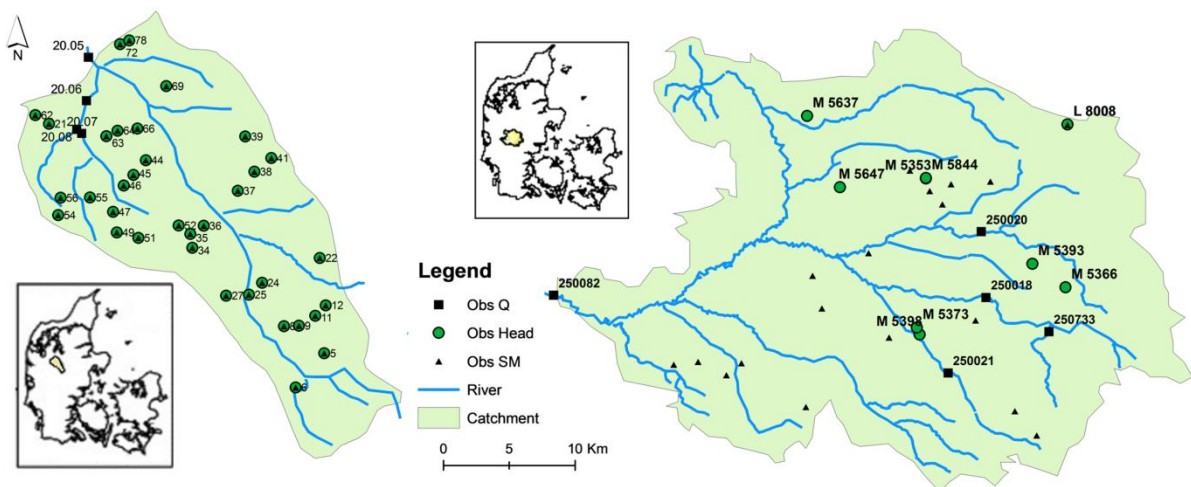

3 **Figure 1 Left: Karup catchment, Right: Ahlergaarde catchment. 'Obs Q', 'Obs Head' and 'Obs SM' represent discharge,**
4 **groundwater head and soil moisture observations respectively used for assimilation.**





**Figure 2 Sketch of localization scheme for soil moisture at a site where soil moisture is measured at 0–5 cm and 20-25 cm (marked by filled black circles). The depths at right represent the numerical layers. The dotted-line ovals indicate the localization areas for each layer, where the cut-off values of covariance function increase quadratically from depth 20 - 25 cm downward.**





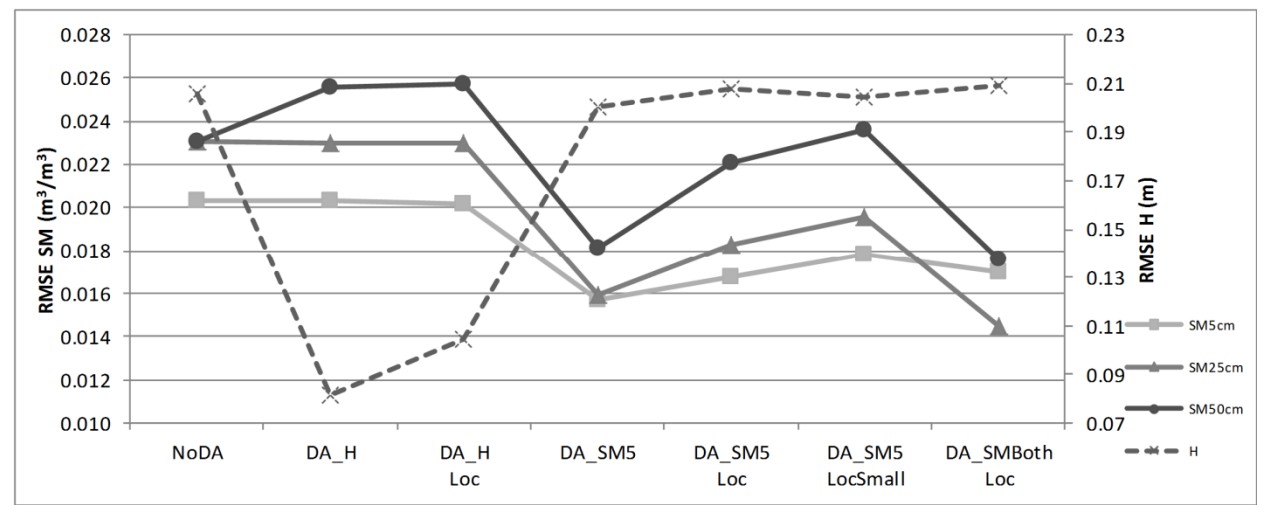

3 **Figure 3 Spatial averaged RMSE of groundwater head and soil moisture at different depths for the analysed univariate**
4 **assimilation experiments of in Karup catchment. Left axis represents soil moisture and right axis head.**



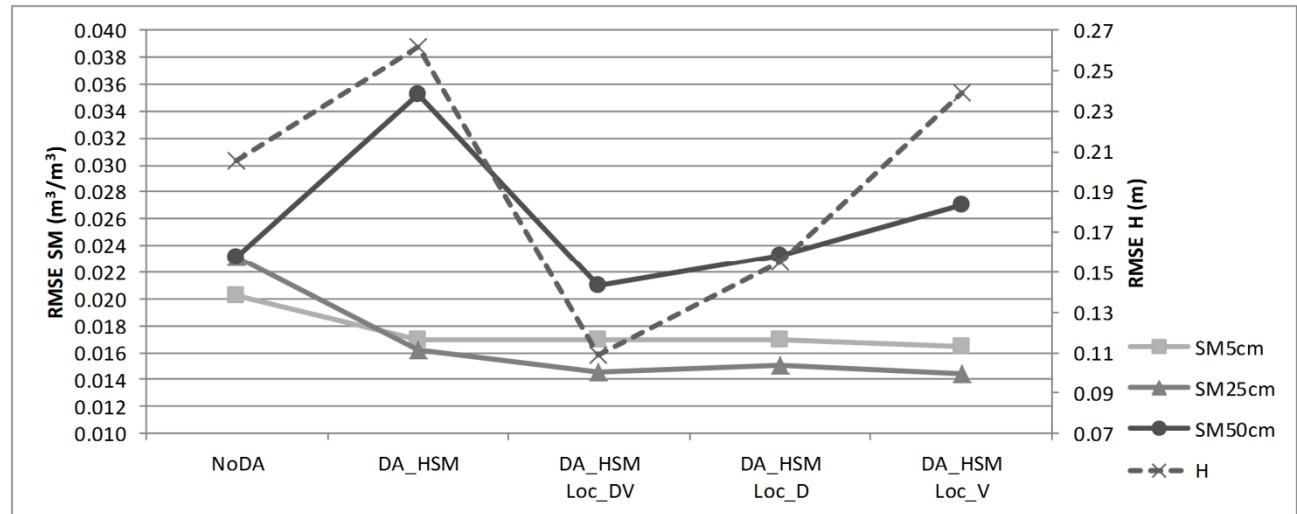

**Figure 4 Spatial averaged RMSE of groundwater head and soil moisture at different depths for each multivariate assimilation experiment in Karup catchment . Left axis represents for soil moisture and right axis head.**





**Figure 5 Results from different experiments in Karup catchment. From top to bottom, the 1st panel shows the average spatial RMSE of groundwater head. The 2nd, 3rd and 4th panels are the average spatial RMSE of soil moisture at 5 cm, 25 cm and 50 cm depths respectively. From left to right, the experiment names are indicated as the horizontal axis lable from the bottom panel. For each experiment except NoDA, the results of three ensemble sizes (30,60 and 90) are represented using different colours as shown in legends.**





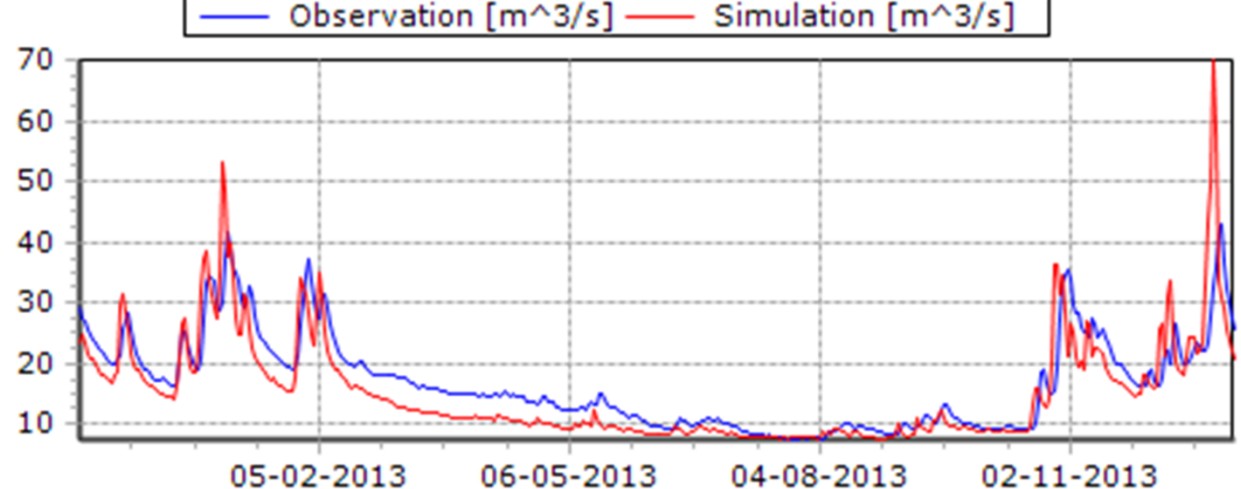

4   **Figure 6 Modelled and observed hydrograph at the outlet of Ahlergaarde catchment in 2013.**





3  **Figure 7 Top: Groundwater head at well M5373. Middle: soil moisture at 2.5cm at site nw1.1. Bottom: soil moisture at 22.5cm**
4  **depth at site nw1.1. The light gray lines (not marked in the legend) are the open-loop ensemble prediction. 'Mean' (single gray**
5  **line) is the ensemble average. 'Deter' (dark line) is the deterministic model. 'Obs' (cross mark) are the observations.**







**Figure 8 Top: groundwater head at well M5373. Middle: soil moisture at 2.5 cm at site nw1.1. Bottom: soil moisture at 22.5 cm**
**depth at site nw1.1. The light gray lines (not in the legend) are ensemble predictions. 'Mean' (single gray line) is the ensemble**
**average. 'Deter' (dark line) is the deterministic model. 'Obs' (cross mark) are the observations. Note the assimilation starts from**
**2012-11-01.**







3   **Figure 9 Spatial RMSE between assimilated and deterministic model in Ahlergaarde catchment : soil moisture at 2.5cm depth**
4   **(upper left) and 22.5cm depth (upper right), groundwater head at layer 4 (lower left) and layer 5(lower right). The observation**
5   **locations at each layer are marked with violet crosses.**



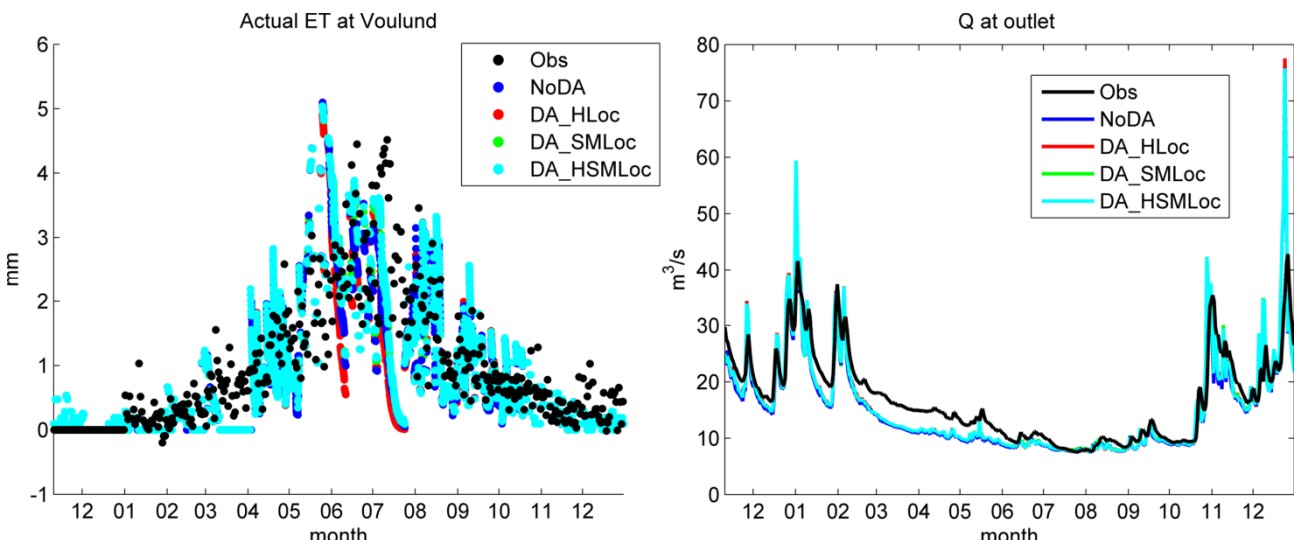

3 **Figure 10 Left: Actual evapotranspiration in each experiment and observed evapotranspiration at Voulund at Ahlergaarde**
4 **catchment. Right: discharge at Ahlergaarde catchment outlet (station 250082) for each experiment and observed discharge.**

