# Peer review of "Multivariate hydrological data assimilation of soil moisture and"

_Hydrology and Earth System Sciences, 2016_

## Referee Comment (RC1) · Anonymous Referee #1 · 9 May 2016

General evaluation:

This paper presents a novel contribution which is the joint assimilation of soil moisture and piezometric head data in integrated hydrological models. The paper investigates different scenarios for univariate and bivariate assimilation, and the role of ensemble size and localization. Especially the role of localization is important and the paper reaches the conclusion that a combination of distance localization and variable localization gives the best results. This is an important conclusion in the context of data assimilation for integrated models. It can be questioned whether for a larger ensemble size the importance of localization would be reduced and variable localization would not be needed. It is however also clear that for many applications a large ensemble size cannot be afforded.

[Figure]

I recommend minor revision for the paper.

Detailed points:

P2, L17-L29: Kurtz et al. (2014), WRR assimilated piezometric heads and groundwater temperatures.

P4, L22-L23: How is the irrigation handled then?

P6, L5-L9: Is this a realistic perturbation of the forcings? Spatial correlation is excluded and therefore the perturbations have less influence than in case larger grouped areas would get either a positive or negative perturbation of the precipitation.

P11, L1: If this would be the case, it would be better not to use localization at all. Was the optimal localization length used, and how was it determined? Was the correlation length of hydraulic conductivity taken into account?

P11, L24-L27: Can this be related to Non-Gaussian distributions as soil moisture is Non-Gaussian distributed? At this point, it would be good to know whether soil moisture or pressure is updated in the data assimilation. MIKE-SHE calculates internally with pressure, so probably pressure was updated in/after the data assimilation procedure. This implies that soil moisture data have to be transformed to pressure for which soil hydraulic properties are needed. In addition, pressure shows in general strongly non-Gaussian distributions, especially under drought conditions. If instead the data assimilation is done in terms of soil moisture, and soil moisture is updated, I wonder how piezometric head is assimilated. For those cases, and the grid cells affected (the grid cell with the groundwater level and the grid cells below the groundwater level), soil moisture could be set equal to porosity. Was this done? It would be good to have some more detail here as this also affects non-linearity/non-Gaussianity in the DA and therefore affects the results.

Section 4.3: Further increase of ensemble size could improve results further. In my opinion, the ensemble size is an unresolved issue.

P13, L14-L15: Is pressure perturbations also transferred to the surface water domain? This could generate the observed stronger perturbations in one of the simulation experiments in the discharge.

P17, L7-L8: Would a coarser model but a much larger ensemble size not be better? The number of grid cells could for example be reduced by a factor of 4 (half of current resolution) and increase the number of ensemble members by a factor of 4.

---

## Referee Comment (RC2) · Anonymous Referee #2 · 26 May 2016

In the manuscript titled "Multivariate hydrological data assimilation of soil moisture and groundwater head" authors have assimilated groundwater head (GW) and soil moisture (SM) observations into Mike She hydrological model using ETKF in a synthetic and a real data assimilation scenarios. Later, these assimilation analyses are validated via comparisons of the accuracy statistics for soil moisture, groundwater level, evapotranspiration, and discharge variables. The study builds on earlier studies that have investigated the ensemble size, localization, and model uncertainty impact on performance of a filter assimilating GW in Mike She model in a synthetic setup (Rasmussen et al., 2015; Zhang et al., 2015). Here authors are assimilating real data instead of synthetic data. Overall, the study is interesting and relevant to HESS Journal. However, the manuscript has many missing details and inconsistencies that prevent acceptance of the study in its current form.

[Figure]

1) Synthetic and real data assimilation experiments are not performed or presented in a consistent way. This prevents results to be meaningfully compared:

1.a) Synthetic studies show "localization degrade analysis for the univariate case when either SM or GW observations are assimilated (Figure 3)" while the multivariate synthetic case and the real data assimilation case shows "improvements in the analysis (Table 3 and Figure 4)". Synthetic studies always reflect the ideal/perfect conditions as they can be fully controlled (particularly the observation errors). I don't see the utility of an application that does not give satisfactory results using synthetic simulations. On the other hand, I am completely puzzled how real data scenario improves the analysis (Table 3) despite synthetic studies fail (Figure 3). Frankly, I would have been more convinced if synthetic results showed improvements while real DA case showed problems (after all real life is not perfect), but not vice versa. This inconsistency should be explained in detail.

1.b) Only the total ET values are given for the synthetic simulations, but not the error statistics of ET (Table 2). They should be given similar to real data assimilation scenario.

1.c) Pick an accuracy statistics (R2; RMSE; or Nash-Sutcliffe efficiency, NSE) and just present it in a consistent way through out the study. Presentation of mixed statistics is very confusing: Table 2 shows R2 and NSE for discharge, Table 3 shows RMSE for SM, Table 4 shows R2 and RMSE for ET while the same table shows R2 and NSE for discharge, Figures 3,4,5,9 shows RMSE. To start with, just give the equations of the error chosen statistics. In general, showing both NSE and RMSE is redundant. NSE ( 1 - sum(X-Y)ˆ2 / sum(X-mu_x)ˆ2 ) can be shown to be equal to (1-RMSEˆ2/variance_x), implying there is a directly relationship between NSE and RMSE: if NSE is high we can expect RMSE to be low and vice versa.. This analytical expectation is also supported by the results given in Tables 2 and4: higher R2 experiments have higher NSE and lower RMSE. Briefly, these three statistics are consistent with each other in representing the accuracy of the variable of interest. Just pick one out of these three statistics

and just show it consistently for all cases (I recommend RMSE in this case, it is up to authors though).

1.d) Table 2 (synthetic data assimilation experiments) is missing "DA_H" and "DA_SM5" scenarios (no localization scenarios). These scenarios are necessary to see whether or not Localization improves or degrades the analysis accuracy. Given one of the major conclusions the authors are making is "Localization does not only provide better results . . .", presentation of these results is very important.

1.e) Figure 3 (synthetic data assimilation experiments) is missing "DA_SMBoth" (no localization scenario). Similar to above, these scenarios are necessary to see whether or not Localization improves or degrades the analysis accuracy.

1.f) Tables 3 and 4 (real data assimilation experiments) are missing "DA_H", "DA_SM", and "DA_HSM" scenarios (no localization scenarios). Similar to above, these scenarios are necessary to see whether or not Localization improves or degrades the analysis accuracy.

2) I see SM simulations/assimilation experiments are not realistic.

2.a) SM observations often have inconsistencies with models (see Reichle and Koster, 2004). For this reason, SM observations are matched to model before they are assimilated. Have authors done something similar? I see authors have matched the means of GW before observations are matched (page 14, line 9), but not the standard deviations or any other statistical property. To start with, do model and observations have similar statistical property? Did authors check the innovations of ETKF? Is it white?

2.b) Many satellite missions (e.g., SMAP) have the goal of retrieving SM observations with 4% error and consistently Mike She model using real data has similar error magnitude. However, the synthetic Mike She model runs do not seem to be realistic with SM errors of 2%.

2.c) Root-zone SM varies much slower than surface SM. As a result, actual root-zone

SM errors have smaller magnitude than surface SM errors (i.e., root-zone RMSE is expected to be smaller, while RMSE values normalized with actual variability of the variable could be different for two layers). In the current study, root-zone SM errors are higher than surface SM errors (Fig. 4). Deeper layer (22.5cm) SM seems more noisy than surface (2.5cm) as shown in Figure 9. Explanation is needed.

2.d) Above points can be clarified very easily using a table showing: mean and standard deviations of observations and model (no assimilation) for both SM (at 5, 22.5, 50cm depths) and GW. Such a table seems necessary to clarify all of above points.

3) Why select ETKF but not EnKF? In atmospheric sciences ETKF could be a viable option because it does not perturb observations (atmospheric models are chaotic, hence such perturbations could be problematic; but hydrological models are not chaotic). Avoiding the perturbation of observations does not seem to be a sufficient justification to use ETKF. Many studies use EnKF, hence it is more relevant to general audience. Briefly, it is better to say "ETKF selection is arbitrary, EnKF could have been selected as well" rather than justifying the ETKF selection via "it avoids the additional perturbation step". In case authors would like to support the selection of ETKF with "it avoids the additional perturbation step" argument, then they should support this claim with a reference (i.e., a study shows this additional perturbation could be problematic in hydrological sciences).

4) Many details are not given in the paper and currently the experiments cannot be replicated by another researcher. Besides many experiment set up decisions looks very arbitrary:

4.a) which datasets are available for the warm up period over Ahlergaarde (i.e., for forcing/parameter)?

4.b) forcing and parameter perturbation statistics for real data assimilation case are completely missing (which forcing variables/parameters are perturbed?, additional/multiplicative noise? mean?, standard deviation?, daily/weekly perturbation?,

etc). How did authors decide about these statistics, justification?

4.c) for the synthetic experiments observations are perturbed using noise with 0.15m and 5% error standard deviation. Why these numbers? Do authors know Decagon 5TE SM sensors have errors of 5% at daily time step? (i.e., SM observations are assimilated at daily time steps even though they are observed every 30min; implying the errors at daily time steps could be much lower than 30 min time step).

4.d) For the real data assimilation scenario, the soil moisture error standard deviation is assumed constant 5%. Why 5%? Is it something the company who produces these Decagon 5TE sensors suggests? Besides, these half hourly observations are averaged into daily values, implying the observation noise if further reduced. Justification for the observation error standard deviation is needed.

4.e) perturbation details in generation of ensembles is missing as well (the study of Zhang et al., 2015 is referred for these perturbations but the experiment details should be given specifically).

4.f) what is the temporal resolution of the model? Daily? It is forced by daily precipitation and reference evaporation (page 4 line 26), bi-hourly GW observations are obtained for real DA case (page 14, line 2), and GW observations interpolated to weekly time steps are assimilated (page 14, line 28) but the temporal resolution of the model (i.e., at what time step the model is run) has never been explicitly mentioned.

4.g) if the model is run at daily time step, then it is not clear how weekly GW observations are assimilated.

4.h) How ET observations are obtained at the stations? What kind of observations are these (eddy covariance? lysimeter? pan evaporation? Reference ET?)? How frequently obtained? No details are given.

4.i) Which data are used as validation? Do authors use the same stations where observations are collected? Perhaps some stations should be reserved and observations

obtained over these stations should not be used in any ways (e.g., assimilation) and should be left purely for validation. After all, if the station has soil moisture probe, then why bother with estimating the SM over that location (this comment is only relevant with direct validation of SM, should not be thought in the framework of SM assimilation to estimate other parameters such as runoff, ET, etc).

4.j) Have you considered assimilating remote sensing-based SM data? Why not?

4.h) "The assimilation performance is evaluated by comparing model output with the actual observations using average RMSE." Now I am puzzled, which observations are assimilated and which observations are used as validation? Are they the same?

5) The study could present more literature review.

5.a) For example Franssen & Kinzelbach (2008) did assimilate GW observations into a hydrological model. Mike She model background/physics (page 4, lines 14-20) could be supported with references as well. ETKF is first introduced by Bishop et al., (2001) not by Sakov et al., (2010); the latter study used EnKF they did not even use ETKF (Page 3, line 16).

5.b) SM and GW time scales are often very different; the impact of precipitation/SM variations could translate to GW variations only after days/weeks/months, depending on the catchment/GW levels/conductivity/etc. On the other hand, the Mike She model simulations have daily time steps (my understanding daily, I could be wrong → authors should clarify this info). If the time scale is very long, it is not immediately clear how SM anomaly for today will give meaningful information about GW anomaly of today. Perhaps the filter (ETKF) should be changed to accommodate past observations? So, sufficient motivation about the utility of SM observations to improve GW simulations using ETKF should be given.

6) "Although the improvements are relatively small, we nevertheless see the benefits in other model process results when improving groundwater head and soil moisture."

[Figure]

It is not clear what is meant with this sentence; do authors imply "we did not find improvement but we think it is still likely in other applications, hence GW and SM should be assimilated together"? After improvements were not found in this study, I think the only comment can be said is "these results should be verified using other models". The interpretation of "seeing the benefits in other model process results" is not can not be made using the results of this study alone.

7) "Localization does not only provide better results . . ." (Page 16, line 25), synthetic results do not support this comment (Figure 3).

8) Abstract requires slight modification: the assimilation method used here "ETKF" has been around for almost 15 years, so this is not the first time it is introduced. Assimilation is not a new method either, has been around for a long time too. So, this current study is just a case study (GW and SM observations are assimilated in a hydrological model). Abstract should be changed to something similar to below:

"Observed groundwater head and soil moisture profiles are assimilated into an integrated hydrological model. The study . . .. model code. Experiments were firstly performed using synthetic data in a catchment of less complexity (the Karup catchment in Denmark), and later performed using real data in a larger and more complex catchment (the Ahlergaarde catchment in Denmark)."  

MINOR:

Page 3, line 15, ". . . performance of assimilating soil moisture" → ". . .performance of a filter assimilating soil moisture".

Page 4, line 4, "Karup catchment is well-studied catchment", in terms of what parameters/variables?

Page 10, line 17, "assimilating soil moisture at 5 cm depth . . .. and 1 m depth". 5 cm refers to the depth of the assimilated observations, how about 1m? Soil moisture states up to 1m depth are updated? Is it what is meant?

[Figure]

Page 10, line 18-19, similar to above (Page 10, line 17).

Page 13, line 21, "spilt" → "split"

Page 16, line 14, "EnKF" → "ETKF"? I believe authors imply "ETKF is a flavor of EnKF" here, but still use of EnKF is confusing since EnKF is not used in this study.

Craig H. Bishop, Brian J. Etherton, and Sharanya J. Majumdar (2001). Adaptive Sampling with the Ensemble Transform Kalman Filter. Part I: Theoretical Aspects. Monthly Weather Review, 129:3, 420-436.

Reichle R. H., Koster R. D. (2004). Bias reduction in short records of satellite soil moisture. Geophysical Research Letters, 31, L19501, pp 1-4.

Hendricks Franssen, H. J., and W. Kinzelbach (2008). Real-time groundwater flow modeling with the Ensemble Kalman Filter: Joint estimation of states and parameters and the filter inbreeding problem, Water Resour. Res., 44, W09408, doi:10.1029/2007WR006505.

———————————————

---

## Referee Comment (RC3) · Anonymous Referee #3 · 30 May 2016

General comments

This paper presents a study of hydrological data assimilation for integrated catchment modeling, using a combination of ensemble Kalman filter and the MIKE SHE model. By assimilating groundwater head and soil moisture data, the Authors investigate the filter performance in various scenarios characterized by different observations assimilated, distance- and variable-based localization, and ensemble size. Two test cases are considered, one with purely synthetic data and one with real (albeit "processed", see specific comments below) observations. The paper is of interest for the readers of HESS and, although the methods are not new, there are some original features, such as the use of distance and variable localization for data assimilation in an integrated hydrological model. However, a number of issues should be addressed before the paper can be considered for publication.

[Figure]

1) Many details are missing and therefore the numerical experiments are not reproducible. In particular, there is no mention whatsoever of the model parameters (e.g., soil properties) and how they were perturbed to generate the initial ensemble of realizations. Precipitation and potential evapotranspiration were also perturbed but no description on how this was done is currently available in the paper. This is especially relevant for the Ahlergaarde catchment, while perhaps some information on the Karup catchment might be available in previous publications by the same research group.

2) The "deterministic" model has been calibrated against observation data for both catchments, yet for the Karup catchment this is not showed neither in figures nor tables, while for the Ahlergaarde catchment only a comparison between observed and simulated discharge at the outlet is included, which is a bit limited, compared to the capabilities of the model. I think it is important to show and briefly discuss the model calibration performance in both cases, maybe also in terms of water table, to build some confidence that the subsequent analyses are realistic.

3) One of the recurrent results of this study is that assimilation of groundwater head does not improve (or even worsen) soil moisture and vice versa, whereas Camporese et al. (Vadose Zone Journal, 2009), in a similar study, showed that EnKF-assimilation of surface soil moisture can improve the saturated zone and assimilation of groundwater head can improve surface soil moisture. This is probably due to the fact that the model used by Camporese et al. (VZJ, 2009) seamlessly solves the entire subsurface domain with the full 3D Richards equation, while here there might be some "disconnection" between the unsaturated domain and the saturated one. This may lead to the risk of physically inconsistent updates and hence the need for variable localization. I believe that a more insightful discussion is required about the relationship between the DA results and how the saturated zone and the unsaturated one are coupled in MIKE SHE, especially for the readers that, like me, are not entirely familiar with how process coupling is done in such model.

Specific comments

Page 1, lines 18-19: improvements obtained in discharge and ET with data assimilation were only marginal, as acknowledged by the Authors later on in the manuscript. I suggest this statement be relaxed.

Page 2, lines 4-6: this is not a general definition of hydrological data assimilation. Please rephrase.

Page 4, lines 22-23: irrigation is not included, yet it is said that the catchment is located in one of the most irrigated areas of Denmark. I think a much stronger justification is warranted than simply "for computational efficiency".

Page 7, line 8: the true state in reality is never known, therefore it is not sure it can be always represented. I suggest this sentence be reformulated.

Page 9, line 15: "appropriate model error" is not sufficient, please give more details.

Page 10, line 23 and Figure 3: I assume RMSE is averaged not only in space but also in time. Please clarify.

Page 12, line 18: does it mean that previous experiments were run with an ensemble size of 60? If so, please clarify. Also, please provide justification for using 60 (e.g., previous sensitivity analysis?).

Page 13, lines 22-26: more details are needed regarding the perturbation of precipitation, evapotranspiration, and parameter values. For instance, how were the variables perturbed? With additional Gaussian noise? And using what variance? Also the model parameters (e.g., soil properties) must be specified.

Page 13, lines 29-30: from the figure it is clear that the model consistently underestimates low flows and overestimates peak flows. Therefore, I do not think that "the model is good at predicting low flow" and suggest this sentence be reformulated. What about performance of the model against other data (e.g., water table, soil moisture)?

Page 14, lines 7-10: this procedure for "adjusting" the observations and make them

closer to the model predictions is a strong assumption and represents a significant limitation of this study. By doing this, what would be a real test case becomes basically another synthetic experiment. If the Authors really think the observations are biased (and it seems they justify this hypothesis based only on the fact that the calibrated model is not able to match the data, which is a weak explanation, in my opinion), why don't they use the framework proposed by themselves in Rasmussen et al., "Data assimilation in integrated hydrological modelling in the presence of observation bias", HESS, 2016? Such an analysis would make the paper more robust and interesting.

Page 14, lines 16-19: this explanation is needed earlier in the manuscript. See previous point concerning Page 4, lines 22-23.

Page 14, lines 19-20: more details are needed here. Which criteria were used to remove unreliable observations from the dataset?

Page 16, line15: data used in this study are not exactly "real" (see previous comment). Please relax this statement.

Page 17, line 4: improvements in discharge and ET are very small, I would say "marginal", instead of "relatively small".

Page 20, Table 2: please remove the column with cumulative ET and replace it with measures of performance of the various DA scenarios, as done with discharge at the outlet.

Page 22, Table 4: first column (cumulative ET) can be removed.

Page 31, Figure 9: not clear what this figure should represent. Here RMSE is computed with respect to the "deterministic" model, which is clearly not the truth, is surely affected by strong uncertainty, and may well be erroneous (as shown by model results in Figure 6). If the Authors want to quantify the spatial distribution of the system corrections made by DA, I suggest replacing RMSE with, e.g., AAD (average absolute deviation) between the DA runs and the deterministic model. Finally, please add labels to x- and

y- coordinate axes.

Technical corrections

All over the manuscript: correct all citations where the Authors are erroneously in parentheses, e.g., (De Lannoy et al., 2007) applied EnKF . . .

Page 4, line 4: replace "relative" with "relatively".

Page 4, line 27: correct "Metrological". In general, small edits of English are required throughout.

Figure 5: please correct labels in the top three panels.

---

## Author Comment (AC1) · 2 Jul 2016

Thank you very much for your time to review our manuscript. All our comments are available in the supplement files in attachment. Best regards

Please also note the supplement to this comment:
http://www.hydrol-earth-syst-sci-discuss.net/hess-2016-126/hess-2016-126-AC1-supplement.zip

---

## Author Response (AR1)

**Reply letter**

Dear Editor,
Thank you for the additional comments which are very helpful. Please find the responses one by one in this reply letter.

1. The comments from Referee 2 are quiet helpful, however, some of them are not well addressed. For example, Question 2.c) and 2.d are misunderstanding. It is normally assumed that higher dynamics of a variable, more difficult to be reproduced and bigger magnitude of errors. Following this logic, the better model performance of depth-layer soil compared to upper-layer soil should be discussed. In addition, Question 4c and Question 7 should be clarified in the revised manuscript.

1) We agree that the comments from Referee#2 are very helpful. Here we did what is suggested in 2.d), in order to address other points. As can be seen in figure 1 below, the surface soil moisture shows larger degree of temporal dynamics in terms of both averaged value and the standard deviation in No assimilation experiment. However, we notice a relatively larger standard deviation in depth-layer (50cm) compared to the surface layer (5cm). This is due to the fact that more soil moisture grid cells are above saturation threshold in deeper layers, which leads to a larger averaged deviation spatially. When the model parameters are perturbed for the 'true' model in the synthetic experiment, the level of saturation especially in depth-layer can be altered, which could lead to a larger RMSE for the depth-layer compared to the surface layer. We agree with the logic that the higher dynamics of a variable, the more difficult it is to reproduce and therefore lead to bigger magnitude of errors. However, when including those inactive soil moisture cells (above saturation), the 50cm layer has relatively larger standard deviation for the entire domain, which make it more difficult to reproduce. This explanation is also added in the revised manuscript (P11 L26-29).

[Figure]

Figure 1 The spatially averaged mean and standard deviation of soil moisture at three depth in NoDA (No assimilation) model for simulation period of 1 year.

The Question 7 is a wrong observation from the reviewer as we also clarified in 1.a. In our study, both synthetic and real cases support this argument. Localization does provide better results compared to the case where no data is assimilated (NoDA).

2. Cross-validation is recommended as the same station data are used as both input and validation. For example, leave 1/3 stations for validation and 2/3 as input.

2) We agree that cross-validation can be adopted to study the DA performance with respect to different observation amount and observation sites. One example can be seen in the referred reference (Zhang et al., 2015). The main focus of this study is to investigate the DA performance with respect to different assimilated variable, which can also be seen as a cross-validation for the integrated hydrological modelling. Overall, in literatures of DA application, it seems that the sites-wise cross validation is not commonly used, not as common as in the model calibration/validation applications. The reason is the fact we usually would like to have the better observations in terms of the quantity, quality and variety assimilated in real DA applications.

3. I think it should be fully discussed (e.g., previous investigations) concerning to "assimilation of groundwater head does not improve soil moisture and vice versa". As is also stated by Referee 3.

3) We agree the result of multivariate assimilation should be more discussed. This is done in the revised manuscript in the last section. We added a long paragraph explaining the saturated and unsaturated zones coupling and the multivariate assimilation result. We believe the readers which are not familiar with how process coupling is done in MIKE SHE can have a better understanding after

reading the latest manuscript. As recommended by Referee #3,  We also added the previous findings ((Camporese et al., 2009)as pointed out by Referee #3) and relevant discussion in the revised manuscript (P17 L11-23).

4. The comment from Referee 3: "Page 17, line 4: improvements in discharge and ET are very small". The suggestion is not addressed.

4) The suggestion is included in the revised manuscript (P18 L4).

5. All the tables and figures should be checked and revised following HESS "Manuscript preparation guidelines for authors".

5) All tables and figures are checked and revised following HESS "Manuscript preparation guidelines for authors" in the revised manuscript.

Best regards,
Donghua Zhang

Camporese, M., Paniconi, C., Putti, M., and Salandin, P.: Comparison of Data Assimilation Techniques for a Coupled Model of Surface and Subsurface Flow, Vadose Zone J., 8, 837-845, 10.2136/vzj2009.0018, 2009.
Zhang, D., Madsen, H., Ridler, M. E., Refsgaard, J. C., and Jensen, K. H.: Impact of uncertainty description on assimilating hydraulic head in the MIKE SHE distributed hydrological model, Adv. Water Resour., 86, Part B, 400-413, http://dx.doi.org/10.1016/j.advwatres.2015.07.018, 2015.